# Association between Microsatellite Instability Status and Peri-Operative Release of Circulating Tumour Cells in Colorectal Cancer

**DOI:** 10.3390/cells9020425

**Published:** 2020-02-12

**Authors:** James W. T. Toh, Stephanie H. Lim, Scott MacKenzie, Paul de Souza, Les Bokey, Pierre Chapuis, Kevin J. Spring

**Affiliations:** 1Medical Oncology, Ingham Institute of Applied Research, School of Medicine, Western Sydney University and SWS Clinical School, UNSW Sydney 2170, NSW, Australia; 2Division of Colorectal Surgery, Department of Surgery, Westmead Hospital, Sydney 2145, Australia; 3Department of Colorectal Surgery, Concord Hospital and Discipline of Surgery, Sydney Medical School, University of Sydney, Sydney 2137, Australia; 4Liverpool Clinical School, Western Sydney University, Sydney 2170, Australia

**Keywords:** circulating tumour cells, colorectal cancer, colorectal surgery, microsatellite instability

## Abstract

Microsatellite instability (MSI) in colorectal cancer (CRC) is a marker of immunogenicity and is associated with an increased abundance of tumour infiltrating lymphocytes (TILs). In this subgroup of colorectal cancer, it is unknown if these characteristics translate into a measurable difference in circulating tumour cell (CTC) release into peripheral circulation. This is the first study to compare MSI status with the prevalence of circulating CTCs in the peri-operative colorectal surgery setting. For this purpose, 20 patients who underwent CRC surgery with curative intent were enrolled in the study, and peripheral venous blood was collected at pre- (t1), intra- (t2), immediately post-operative (t3), and 14–16 h post-operative (t4) time points. Of these, one patient was excluded due to insufficient blood sample. CTCs were isolated from 19 patients using the Isoflux^TM^ system, and the data were analysed using the STATA statistical package. CTC number was presented as the mean values, and comparisons were made using the Student *t*-test. There was a trend toward increased CTC presence in the MSI-high (H) CRC group, but this was not statistically significant. In addition, a Poisson regression was performed adjusting for stage (I-IV). This demonstrated no significant difference between the two MSI groups for pre-operative time point t1. However, time points t2, t3, and t4 were associated with increased CTC presence for MSI-H CRCs. In conclusion, there was a trend toward increased CTC release pre-, intra-, and post-operatively in MSI-H CRCs, but this was only statistically significant intra-operatively. When adjusting for stage, MSI-H was associated with an increase in CTC numbers intra-operatively and post-operatively, but not pre-operatively.

## 1. Introduction

Biomarkers in colorectal cancer (CRC) have had limited success in clinical application to date, but microsatellite instability (MSI) status is emerging as a biomarker of clinical relevance. It is known that CRC exhibiting high level MSI (MSI-H) is associated with increased tumour infiltrating lymphocytes (TILs) and is a marker of immunogenicity [1,2,3]. MSI-H CRCs are less likely to disseminate due to TILs as a protective factor, yet a double-edged sword exists in that MSI-H CRCs have more mutations and are associated with more adverse pathological features. However, what is not known is whether the abundance of TILs decreases the risk of tumour dissemination by reducing the release of circulating tumour cells (CTCs), which have metastatic potential.

Microsatellites are short tracts of repetitive sequence (1–6 base pairs or more that are generally repeated between 5 and 50 times) found disseminated throughout the genome. Due to the repetitive nature of microsatellites, these regions are prone to change (instability) during replication. In MSI-H CRC, the resultant microsatellite alterations result in frameshifts that truncate proteins and may lead to inactivation of affected coding regions. Usually, microsatellite alterations are sensed by mismatch repair (MMR) genes that act like spellcheckers or DNA damage sensors, which detect mutations and signal for repair or apoptosis. When there is a loss of DNA damage sensors, either through genetic or epigenetic inactivation of MMR genes, this leads to loss of appropriate signalling and an accumulation of genetic mutations. In clinical practice, MSI-H occurs in 10–15% of colorectal cancers and is defined by IHC staining demonstrating MMR deficiency (MMRD).

The serrated neoplastic pathway is one of the two sporadic pathways that result in MSI-H cancers, the other classical pathway being the adenoma-carcinoma pathway involving chromosomal instability (CIN) and that results in microsatellite stable (MSS) cancers. Interestingly, it appears that MSI-H colorectal cancers are less likely to progress to stage IV disease compared to their MSS counterpart [4,5]. However, it is unclear if the biology of this observation is associated with decreased release of CTC. The hypothesis we sought to test was that the abundance of TILs in MSI-H CRCs may reduce the release of CTCs and, by doing so, protect against the risk of dissemination.

CTCs were first identified by Dr. Thomas Ashworth in 1869 [6]. Under the microscope, it was observed that “cells identical with those of the cancer itself” were present in the blood of a man with metastatic cancer. Since then, CTCs have been shown to be both a predictive and prognostic biomarker, but have remained in the research domain rather than clinical application due to cost. Evaluating circulating tumour DNA (ctDNA) has also shown great utility as a diagnostic approach for cancer management. Instead of identifying cancer cells (CTCs) in the bloodstream, identifying ctDNA depends on DNA released into the bloodstream from the tumour cell nucleus as it dies and is replaced by new cancer cells. A recent study by Tie et al. investigated ctDNA in stage II colon cancer to detect patients at high risk of recurrence. In that study, they also assessed the association between post-operative ctDNA status and conventional high-risk clinicopathological factors, but were not able to show an association, albeit that the majority of patients in the study were ctDNA negative [7].

With the call for universal MSI testing in CRC, it is important to understand the immunobiology of MSI to understand its clinical implications and its role in guiding prognosis and adjuvant therapy. It is known that MSI is associated with TILs [8]. However, it is not yet known if MSI status affects the release of CTC. It is not clear if patients with abundant TILs have a reduction of CTC count and whether this is stage dependent. This pilot study is the first to investigate CTC count in elective colorectal surgery and to analyse possible differences in the pre-operative, intra-operative, and post-operative stage of treatment and correlate this with MSI status.

## 2. Materials and Methods

### 2.1. Patients and Blood Samples

Twenty patients undergoing elective laparoscopic or open colorectal surgery at either Liverpool or Westmead Hospitals were enrolled in the study approved by the South Western Sydney Local Health District Ethics (Ref: HREC/13/LPOOL/158). All patients gave informed written consent for blood collection and CTC analysis. Peripheral venous blood was collected at four time points: (t1) pre-operative blood collection in the anaesthetic bay of operating theatres; (t2) intra-operatively after mobilisation of bowel was completed; (t3) at time of completion of surgery; and (t4) fourteen-sixteen hours post-operatively. One patient had insufficient blood volume collected and was excluded from analysis.

### 2.2. CTC Enrichment and Enumeration

Quantification of CTCs was performed using the IsoFlux^TM^ instrument (Fluxion Biosciences Inc, Alameda, CA, USA). Peripheral venous blood was collected into 9 mL anti-coagulant K_2_EDTA tubes (Vacuette 455036) and processed within 24 h of collection in accordance with the Isoflux protocol using the CTC enrichment (910-0091) and enumeration (910-0093) kits supplied by the manufacturer. Briefly, immuno-magnetic EpCAM linked beads was used to capture CTCs, and after processing through the Isoflux instrument, CTCs were identified by immune staining using anti-cytokeratin (CK-7, -8, -18, and -19), Hoechst 33342 dye, and anti-CD45. After transferring each sample to 24 well SensoPlates^TM^ (Cat. No. 662892, Geriner Bio-One, GmbH, Kremsminster, Austria) and applying coverslips, each sample was scanned and visualized using a 10× objective on a fluorescence Olympus IX71 inverted microscope. Putative CTCs were defined as CK^+^, DAPI^+^, and CD45^−^, nucleated and morphologically intact cells.

### 2.3. Clinical and Histopathological Data

Carcinoembryonic antigen (CEA), tumour infiltrating lymphocytes (TILs), microsatellite instability (MSI), and BRAF status were recorded. TILs were reported by the pathologist as present when there were more than 5 intraepithelial lymphocytes/100 epithelial cells (assessed on minimum three high power (×400) fields) [8]. MSI and BRAF status was tested by immunohistochemistry.

Data on patient demographics, histopathological features of the tumour, and CTC count at four time periods were collected. Certified pathologists examined the tissue biopsy specimens post-operatively and provided the histopathology diagnosis. All patients had follow-up at one year with disease-free survival (DFS) being the main outcome.

### 2.4. Statistical Analysis

Analysis was performed on STATA (Stata MP, Version 15; StataCorp LP). The Student *t*-test was used to compare between groups, and a Poisson regression was used to adjust for stage. The Student *t*-test was used instead of the Wilcoxon–Mann–Whitney U-test and the Kruskal–Wallis tests as these non-parametric tests are better used to compare medians, whereas the Student *t*-test provides a better assessment of means.

## 3. Results

### 3.1. Clinical and Surgical Characteristics

In total, 80 samples from 20 patients with colorectal cancer who underwent elective open or laparoscopic colorectal surgery with curative intent were recruited for this study. However, four samples from one patient had insufficient blood volume collected, and this patient was excluded from analysis. CTC isolation and enrichment were performed using the IsoFlux^TM^ system. Of the nineteen patients who had CTCs enumerated, two patients had high-grade dysplasia without malignancy (these were 30 × 33 × 23 mm and 57 × 50 × 55 mm villous adenomas). Of the remaining 17 patients, 3 (17.6%) were stage I, 6 (35.3%) stage II, 7 (41.2%) stage III, and 1 (5.9%) stage IV. Nine (52.9%) were right-sided (caecal (*n* = 4), ascending colon (*n* = 4), and transverse colon (*n* = 1)), the rest (41.2%) were left-sided (rectum (*n* = 4), rectosigmoid (*n* = 1), sigmoid (*n* = 3)). The histopathology of all seventeen patients was adenocarcinoma. Further, two of the four patients with rectal cancer had neoadjuvant chemoradiotherapy. Patient demographic, clinical, and surgical characteristics are summarized in Table 1, and the CTC yield for each patient at the different time points are in Appendix A.

### 3.2. CTC Yield in All Patients 

First, we looked at the number of CTCs enumerated for all 19 patients (Table 1 and Appendix A. CTC number was presented as the mean values and comparisons made using the Student *t*-test. CTCs were enumerated for 19 patients: microsatellite stable (MSS) (*n* = 15); microsatellite unstable (MSI-H) (*n* = 4), respectively. A Student *t*-test was used to test the difference in CTC number between MSS and MSI-H CRCs, respectively, at the four different time points: t1 (8.2 vs. 12.5, *p* = 0.6191); t2 (23.7 vs. 37.8, *p* = 0.5893); t3 (9.3 vs. 12.3, *p* = 0.7798); and t4 (8.1 vs. 18.8, *p* = 0.3696). It was apparent that at each of these time points, there was no significant difference between MSS and MSI-H patient groups.

### 3.3. CTC Yield in Cancer Patients Only

Excluding the two patients with villous adenoma and high-grade dysplasia, there was a trend towards higher CTC number for MSI-H CRC, but this was not statistically significant between the two groups: t1 (7.9 vs. 12.5, *p* = 0.6191); t2 (22.2 vs. 37.8, *p* = 0.5893); t3 (8.7 vs. 12.3, *p* = 0.7798); t4 (8.3 vs. 18.8, *p* = 0.3696). In addition, a Poisson regression was performed adjusting for stage (I-IV). This demonstrated no significant difference between the two MSI groups for t1. However, t2, t3, and t4 were all associated with an increase in CTC number for MSI-H CRCs.

### 3.4. MSI Status and CTC Number by Stage of CRC

For this analysis, there were no MSI-H patients in the stage I and IV groups; however, there were two MSI-H (caecum and ascending colon) patients in the stage II group; whereas there were four stage II MSS CRC (caecum, rectosigmoid and two rectum) patients. For the stage II patients, there was a significant spike at the t2 timepoint for the MSI-H group (Figure 1; Panel A and Panel B). However, the sample size was too small to perform reliable statistical analysis. For stage III patients, there appeared to be a trend towards higher CTC count for the MSI-H group, but this was not statistically significant: t1 (13.2 vs. 14.5, *p* = 0.9540); t2 (15.2 vs. 23.5, *p* = 0.7700); t3 (3.6 vs. 20, *p* = 0.1981); t4 (3.6 vs. 29, *p* = 0.1589); Figure 1; C and D. When combining data for all stage I-III patients, there was a statistically significant spike in t2 count in the MSI-H group. There was also a trend toward higher CTC number for t1, t3, and t4 in the MSI-H group, but this was not statistically significant (Figure 1; E and F): t1 (7.5 vs. 12.5, *p* = 0.6027); t2 (8.25 vs. 37.75, *p* = 0.0328); t3 (2.5 vs. 12.25, *p* = 0.0878); t4 (3.67 vs. 18.75, *p* = 0.0604).

There was only one patient with stage IV colon cancer. This patient had a right-sided cancer that was MSS and had high CTCs that were persistently elevated with 13 CTCs detected at the pre-operative time point (t1), which increased to 189 during surgery and then remained high post-operatively with 83 and 65 CTCs detected, respectively (Figure 2).

### 3.5. MSI Status and CTC by Side (Right vs. Left) Stage I-III Colon Cancer

There were no left-sided MSI-H CRCs in this study, so it was not possible to compare between MSI-H and MSS for left-sided colon and rectal cancer. However, there were four MSI-H and five MSS right-sided colon cancers. One MSS was excluded from analysis as it was a stage IV CRC. There was no statistically significant difference in CTC number for right-sided colon cancer by MSI status, but there was a trend for increased CTC number at t2, t3, and t4 time points (t1 (20 vs. 12.5, *p* = 0.6375); t2 (22.75 vs. 37.75, *p* = 0.5001); t3 (6.5 vs. 12.25, *p* = 0.5677); t4 (6 vs. 18.75, *p* = 0.0942); Figure 3, Panels A and B).

There were eight left-sided CRCs in this study, of which none exhibited MSI. Two patients received neoadjuvant therapy, whereas six patients did not receive neoadjuvant therapy. There were four rectal cancers, one rectosigmoid and three sigmoid cancers. The CTC number overall was low in this subgroup; however, it appeared there was a difference between those who received neoadjuvant therapy and those who did not (Figure 3, Panels C and D). Reliable statistical analysis was not performed due to the small sample size.

### 3.6. Poisson Regression Model with Post-Estimation Marginal Fundamental Analysis

A Poisson regression model was run with MSI and stage as independent variables. A post-estimation marginal means and marginal effects fundamental analysis was performed. This showed no difference in CTC number at the t1 time point, but a statistically significant difference in CTC number at the t2, t3, and t4 time points with stage as a covariate: t1: 7.96 (6.42–9.50) vs. 12.29 (8.88–15.71); t2: (20.99 (18.55–23.43) vs. 46.01 (38.18–53.83); t3: 7.72 (6.28–9.16) vs. 20.69 (13.88–27.49); t4: 8.03 (6.51–9.54) vs. 21.91 (16.62–27.18). In addition, a Poisson regression model was run for stage of CRC, and a post-estimation marginal fundamental analysis was performed. This showed a significant difference in CTC number at the t1, t2, t3, and t4 time points between stages independent of MSI status (Figure 4). Preoperatively, the CTC number at the t1 time point for stage I was 3.99 (2.48–5.49); for stage II 6.78 (5.33–8.22); for stage III 11.53 (9.58–13.48); for stage IV 19.61 (13.63–25.59).

### 3.7. Non-Parametric Wilcoxon–Mann–Whitney U-test

When comparing the median CTC number between MSS and MSI-H CRCs, there was again a trend toward increased CTC number with MSI-H CRC, but this was only statistically significant at the t2 time point when comparing stage I-III MSS and MSI-H CRC.

## 4. Discussion

There is an abundance of clinical data reporting on the association between MSI status and prognosis in CRC showing that MSI-H may be associated with better prognosis [3,4]. The evidence in the literature shows that CTCs may be important in prognostication of colon cancer [5] in predicting dissemination [6], overall and disease-free survival [7], and lymph node involvement [8,9]. Higher levels of CTCs have been associated with worse outcome and may predict for poor disease-free survival [10,11,12]. Most existing studies assessing the relationship between MSI and prognosis in CRC have been clinical studies or histopathological studies. We took a fundamentally different approach to instead examine the immune-biological characteristics of CRC based on MSI status.

Our study confirmed that CTC measurements correlated with dissemination and stage of disease. There was a statistically significant difference in CTC number with stage (I-IV) and across all time points (preoperatively, intraoperatively, and postoperatively, as shown in Figure 4). However, notably, our study showed that MSI-H CRCs (which have been reported to be immunogenic and associated with enhanced survival) [3,4] were associated with increased peri-operative release of CTCs (Figure 1). Further, both the mean and median CTC count at all time points were higher in the MSI-H group compared to the MSS group. Our hypothesis that increased TILs would decrease peri-operative release of CTCs in MSI-H CRCs was not supported by the data.

While most of the analyses performed were not statistically significant, overall, there was a trend towards increased CTC number at all time points (t1, t2, t3, and t4) in the MSI-H group. When analysing stage II CRC and stage I-III CRC, there was a statistically significant increase in the CTC number for the t2 time point: 3.5 vs. 52, *p* = 0.0005 and 8.25 vs. 37.75, *p* = 0.0328, respectively, in the MSI-H group. For all other comparisons, there was increased CTC number in the MSI-H group, but it was not statistically significant. A Poisson regression was performed to adjust for stage (I-IV). This demonstrated no significant difference between the two MSI groups for the t1 time point. However, the t2, t3, and t4 time points were all associated with increased CTC number for MSI-H CRCs.

The literature on CTCs has shown that a measurement of more than three CTCs per 7.5 mL peripheral blood may be associated with poor survival, although some studies suggest 1–2 CTCs per 7.5 mL may also be associated with a worse outcome [13]. Furthermore, higher post-operative CTC numbers may be associated with a higher risk of recurrence [14], whereas improvements in post-operative CTC number from pre-operative baselines has been associated with better survival [15]. In this study, the median CTC number for the t1, t2, t3, and t4 time points for MSS CRC was one, whereas the median for MSI-H CRC was significantly >3 in all the corresponding time points. However, with data in the literature showing enhanced survival in MSI-H CRC, the high CTC numbers for MSI-H CRCs and low CTC numbers for MSS CRCs found in this study did not correlate with the clinicopathological data reporting on survival existing in the literature. On the other hand, this study may have shown that the cut-off for prognostication of CRC by CTC measurement may actually be influenced by MSI status and that there should not be a single cut-off, but the benchmark may depend on characteristics such as MSI. One hypothesis was that CTCs released into the bloodstream of patients with MSI-H CRCs usually remain microsatellite unstable [16], which maintains an enhanced immunogenic response from circulating lymphocytes in the bloodstream [17]. Thus, its presence may not represent the same risk of metastases as CTCs associated with MSS CRCs [18,19].

What about the protective effect of MSI? The immunogenicity of MSI is believed to be associated with the presence of TILs. Collinearity between MSI-H status and TILs has been shown in clinicopathological data, including the data from our own cohort study [20], and has been traditionally associated with a better prognosis [3,4]. It is believed that the survival and clinical benefits of MSI may be due to its immunogenicity, with MSI associated with increased TILs [2,21,22,23,24] and TILs associated with a better prognosis [2,25,26,27,28,29,30,31,32,33,34]. MSI-H CRCs may also be associated not only with an abundance of intra-tumoral TILs, but also with a higher density of associated cytotoxic, helper, and regulatory T lymphocytes in peripheral blood [17], as well as increased activity in the bone marrow [35]. Studies have shown that both the immunogenic TILs response at the tumour site, as well as the circulatory system are believed to be associated with a decreased risk of lymph node metastases [19,36] and distant metastases [18] in MSI CRCs. The assumption, hence, would be that MSI-H CRCs should be associated with decreased CTC dissemination into the blood.

However, this study showed that the immuno-biology of MSI-H CRCs is more complex than this. As is well established in the literature, MSI-H CRCs are also associated with poor differentiation [19,37], are larger and more likely to be mucinous [38,39,40], as well as being higher grade tumours with a greater mutational burden and the mucinous phenotype more likely to be associated with signet cells. It is also believed that MSI-H CRCs are associated with a high number of frameshift mutation peptides (FSPs) [41,42] when compared to MSS CRCs [43,44]. The MSI hyper-mutational state is also thought to be the reason for poor differentiation and other adverse pathological features of the tumour. The higher grade of tumour associated with hyper-mutational state may be the reason for the increased peri-operative release of CTCs. On Poisson regression analysis, our study showed no difference in pre-operative (t1) CTC numbers, but a statistically significant difference in intra-operative (t2 and t3) and post-operative (t4) release of CTCs. From the literature, immunogenic TILs may reduce the risk of dissemination, but this study showed that they did not do so by preventing the release of CTCs. We currently do not have long term survival and recurrence data, nor CTC measurements outside of the 24 h window peri-operatively, but a study looking at recurrence, survival, and CTC number at 6 months, 1 year, 3 year, and 5 years would be interesting and would examine the role that CTCs may play in the post-operative management of CRCs including whether this may be used to predict recurrence or survival accurately.

Notably, this pilot study showed results that warrant further investigations. The increased CTC positivity seen in MSI-H CRCs is a significant point of difference from the currently understood immune-biological mechanisms associated with MSI-H CRCs. In practice, this means that the benchmark cut-off points for CTC enumeration may be influenced by tumour characteristic, and future clinical applications of CTC in CRC management may need to take this into consideration. Another consideration is that the improved survival associated with immunogenic MSI-H CRCs may not be as profound as once believed, and the increased peri-operative release of CTCs in MSI-H CRCs may be revisited with future studies with a larger patient cohort. This could prove a useful follow up of the suggestions made by the European Society of Medical Oncology (ESMO) guidelines that patients with stage II colon cancer with MSI are at very low risk of recurrence and unlikely to benefit from chemotherapy.

In this study, the overall patient number (*n* = 20) was low and a limitation for insightful statistical analysis. There was significant heterogeneity within the study with two patients receiving neoadjuvant chemoradiotherapy, no direct comparisons available for stage I and IV (no MSI-H CRC in these subgroups), and the inclusion of right colon, left colon, and rectal cancers. The incidence of MSI in CRC is 10–15%. In our cohort, there were only four MSI-H CRCs, of which two were stage II and two were stage III, and the majority of patients were MSS CRCs, being a limitation of our data. From a tumour biology perspective, the main implication of our findings is that immunogenic MSI-H CRCs did not protect against release of CTCs, and the protective effect against metastatic disease was not by reducing CTCs. Further, the main clinical implication of this study involves the utility of CTCs for monitoring and surveillance with potentially differential baseline levels of CTCs associated with the two subtypes of CRC.

## 5. Conclusions

There was a trend toward increased CTC release pre-, intra-, and post-operatively in MSI-H CRCs, but this was only statistically significant intra-operatively. When adjusting for stage, MSI-H was associated with an increase in CTC number intra-operatively and post-operatively, but not pre-operatively. This dataset was limited, and further studies are required. Finally, these data suggested that immunogenic MSI-H CRCs did not suppress CTC release and that different reference ranges may be required for CTC enumeration of MSI-H and MSS CRC.

## Figures and Tables

**Figure 1 cells-09-00425-f001:**
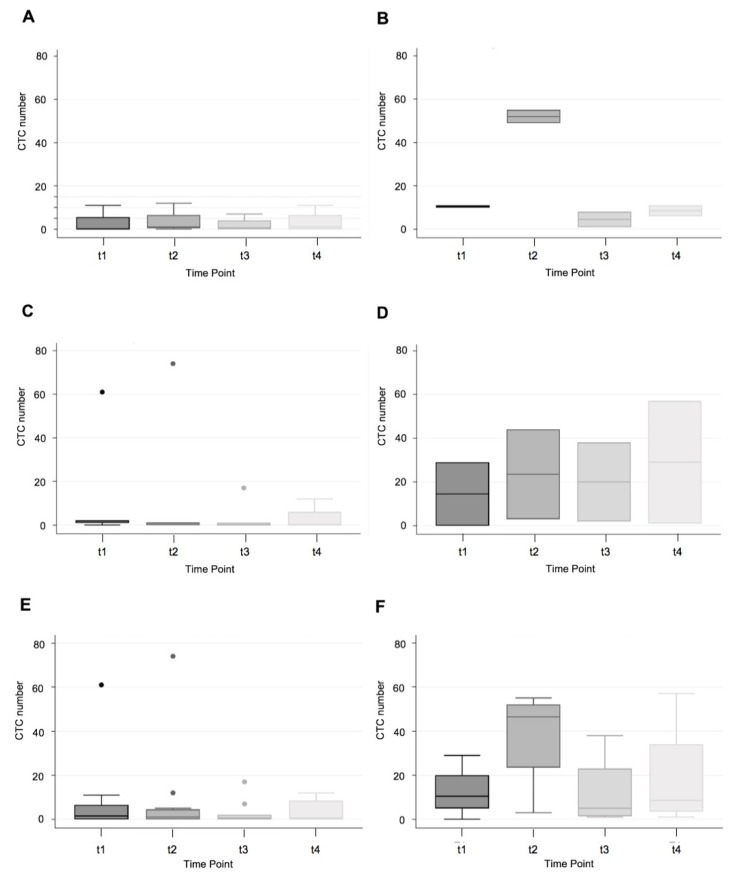
CTC number for stage II (Panels **A** and **B**), III (**C** and **D**) and I-III (Panels **E** and **F**) MSS (left) vs. MSI-H (right) colorectal cancer at different sample time points: t1 (pre-operative), t2 (intra-operative), t3 (immediate post-operative), t4 (14–16 h post-operative).

**Figure 2 cells-09-00425-f002:**
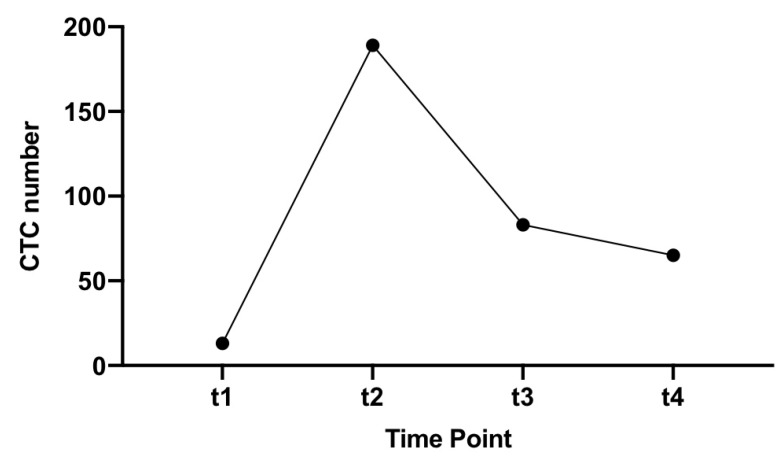
CTC number at different time points for patient S16 (stage IV, MSS CRC).

**Figure 3 cells-09-00425-f003:**
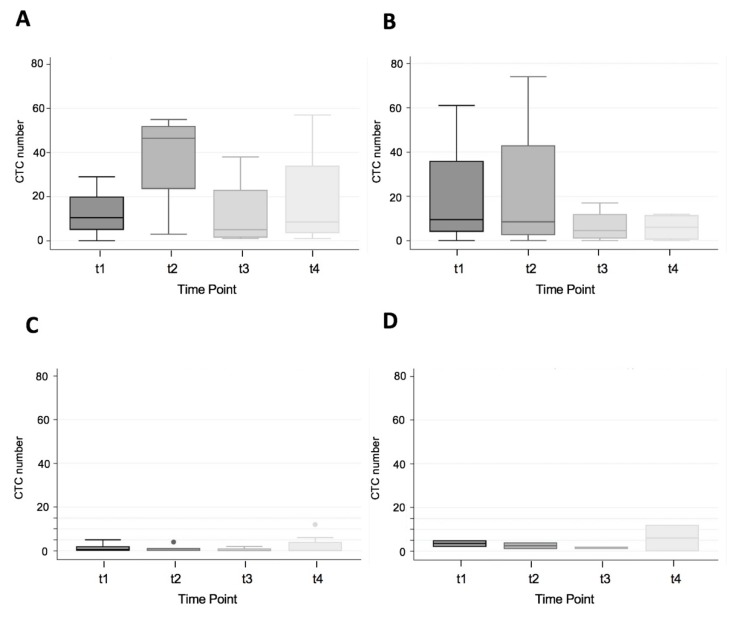
CTC number and MSI status for right-sided stage I-III colon cancer, (Panel **A**) MSS, (Panel **B**) MSI-H and CTC number for left-sided colorectal cancer, no neoadjuvant (Panel **C**), and neoadjuvant therapy (Panel **D**) at sample time points t1 (pre-operative), t2 (Intra-operative), t3 (immediate post-operative), t4 (14–16 h post-operative).

**Figure 4 cells-09-00425-f004:**
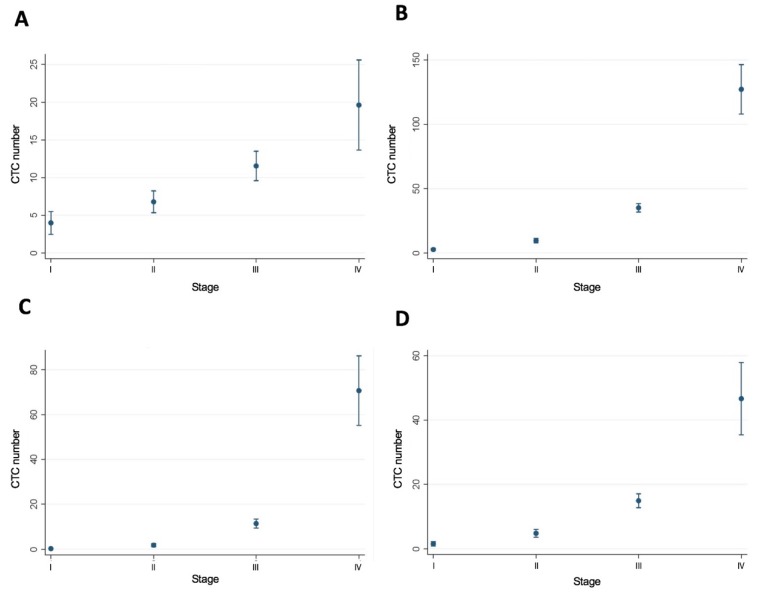
Poisson regression model with post-estimation marginal fundamental analysis of stage and CTC number at sample time points t1 and t2 (Panels **A** and **B**) and t3 and t4 (Panels **C** and **D**).

**Table 1 cells-09-00425-t001:** Patient demographic, clinical, and surgical characteristics and circulating tumour cell (CTC) number. MSI-H, microsatellite instability-high; MSS, microsatellite stable.

Patient Characteristics	Microsatellite Status	
	MSI-H	MSS
Patient number	4	13
Age	85.5 (54–86)	66 (44–86)
Female:male	3:1	6:7
Right colon	4 (100%)	5 (38.5%)
Left colon	0	3 (23.1%)
Rectal/rectosigmoid	0	5 (38.5%)
Grade		
High	3 (75%)	1 (8.3%)
Moderate	1 (25%)	10 (83.3%)
Low	0	1 (8.3%)
BRAF mutant:wild-type	3:1	N/A
Stage		
I	0	3 (23.1%)
II	2 (50%)	4 (30.8%)
III	2 (50%)	5 (38.5%)
IV	0	1 (7.7%)
CTC number		
t1	10.5 (0–29)	1 (0–61)
t2	52 (44–189)	1 (0–74)
t3	23 (1–83)	1 (0–17)
t4	34 (6–65)	1 (0–12)

Continuous data shown as the mean with the range; count data presented as the frequencies and percentages. Three patients are not included in the Table as the histopathology was villous adenoma, with high-grade dysplasia for two patients, and one patient had insufficient samples.

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
