# Peer review of "Association between Microsatellite Instability Status and Peri-Operative Release of Circulating Tumour Cells in Colorectal Cancer"

_cells, 2020, doi:10.3390/cells9020425_

Round 1

Reviewer 1 Report

The manuscript for review entitled “Association between Microsatellite Instability (MSI) Status and Perioperative Release of Circulating Tumour Cells (CTC) in Colorectal Cancer” by Toh et al definitely brings some new information about MSI status affects the release of CTC in Colorectal Cancer. The authors showed the increasing trend toward CTC release pre-, intra- and post-operatively in MSI-H CRCs, but only statistically significant intra-operatively. Despite the general interest in the topic, the work presented in the manuscript is interesting. In my view, the presented data and study may have real utility. My comments are as follows:

The rationale of the study is not clear how the microsatellite status is linked to release of CTC in Colorectal Cancer? How the association of MSI is important in progression in Colorectal Cancer? Please explain. The authors should include the background information about the microsatellite instability in the introduction. How microsatellite status is important in Colorectal Cancer? Please explain MSI and MSS. These information would be helpful to understand for general readers. How the authors decided 20 patients for this study? Why the authors didn’t measure TILs in the study? The authors found an increasing trends toward CTC release pre-, intra- and post-operatively in MSI-H CRCs. What is the implication of this finding? The authors should represent the CTC yield result in the tabular form. It will be helpful to understand the results clearly. The CTC numbers in the stage IV patient (Fig 2) is not conclusive or reliable as it was based on only one patient. The authors should represent the figures clearly. Please correct the legends of Figure 1.   Please define MSI-H? Line 186….. there is no fig 5

Author Response

The manuscript has been extensively edited for typographic and other errors and all changes have been indicated using track changes. The reviewers comments have been addressed below.

REVIEWER 1

The manuscript for review entitled “Association between Microsatellite Instability (MSI) Status and Perioperative Release of Circulating Tumour Cells (CTC) in Colorectal Cancer” by Toh et al definitely brings some new information about MSI status affects the release of CTC in Colorectal Cancer. The authors showed the increasing trend toward CTC release pre-, intra- and post-operatively in MSI-H CRCs, but only statistically significant intra-operatively. Despite the general interest in the topic, the work presented in the manuscript is interesting. In my view, the presented data and study may have real utility. My comments are as follows:

The rationale of the study is not clear how the microsatellite status is linked to release of CTC in Colorectal Cancer? How the association of MSI is important in progression in Colorectal Cancer? Please explain.

We have reworded the first paragraph in the Introduction to better explain the rationale of the study (lines 55-65) and in paragraph 2 (lines 79-81) and now reads:

“Biomarkers in colorectal cancer (CRC) have had limited success in clinical application to date, but microsatellite instability (MSI) status is emerging as a biomarker of clinical relevance. It is known that CRC exhibiting high level MSI (MSI-H) are associated with increased tumour infiltrating lymphocytes (TILs) and is a marker of immunogenicity [1-3]. MSI-H CRCs are less likely to disseminate due to TILs as a protective factor, yet a double-edged sword exists in that MSI-H CRCs have more mutations and are associated with more adverse pathological features. However, what is not known is whether the abundance of TILs decreases the risk of tumour dissemination by reducing the release of circulating tumour cells (CTCs) which have metastatic potential.”

“However, it is unclear if the biology of this observation is associated with decreased release of CTC. The hypothesis we sought to test was that the abundance of TILs in MSI-H CRCs may reduce the release of CTCs and by doing so protect against the risk of dissemination.”

The authors should include the background information about the microsatellite instability in the introduction. How microsatellite status is important in Colorectal Cancer? Please explain MSI and MSS. These information would be helpful to understand for general readers.

It is widely believed that MSI-H CRCs are immunogenic with increased TILs (1-5) with several meta-analyses demonstrating enhanced survival with MSI-H CRCs (6, 7). The major protective effect of MSI-H CRCs believed to be associated with TILS decreasing the risk of lymph node metastases(8, 9) and distant metastases(10). MSS CRCs do not have abundant TILs and are at a higher risk of dissemination.

We have reworded paragraph 2 and 3  in the Introduction to better explain MSI as well as MSI-H and MSS CRCs (lines 64-77) and now reads:

Microsatellites are short tracts of repetitive sequence (1-6 base pairs or more that are generally repeated between 5-50 times) found disseminated throughout the genome. Due to the repetitive nature of microsatellites, these regions are prone to change (instability) during replication. In MSI-H CRC the resultant microsatellite alterations result in frameshifts that truncate proteins, and may lead to inactivation of affected coding regions. Usually microsatellite alterations are sensed by mismatch repair (MMR) genes that act like spellcheckers or DNA damage sensors, which detect mutations and signal for repair or apoptosis. When there is a loss of DNA damage sensors - either through genetic or epigenetic inactivation of MMR genes, this leads to loss of appropriate signaling and an accumulation of genetic mutations. In clinical practice, MSI-H occurs in 10-15% of colorectal cancers and is defined by IHC staining demonstrating MMR deficiency (MMRD).

The serrated neoplastic pathway is one of the two sporadic pathways that result in MSI-H cancers, the other classical pathway is the adenoma-carcinoma pathway involving chromosomal instability (CIN) and results in microsatellite stable (MSS) cancers.”

How the authors decided 20 patients for this study?

The twenty patients were empirically chosen as a sufficient cohort size to explore the hypothesis as an initial pilot study.

Why the authors didn’t measure TILs in the study?

We have previously shown collinearity between MSI-H status and TILs in clinicopathological data, and this data was included in our own cohort study (11). TILs was measured as present vs. absent/inconspicuous/scant, but was not included in this study as performing a subset analysis of MSI status based on TILs status was not of value due to the small numbers. Also MSI is a marker for immunogenicity and is strongly associated with TILs and as such we felt looking at TILs as a separate parameter did not add sufficiently to the study as association with MSI status was the main focus.

The authors found an increasing trends toward CTC release pre-, intra- and post-operatively in MSI-H CRCs. What is the implication of this finding?

We have added the following sentence in the Discussion (lines 595-599) and Conclusions (lines 604-606) and now reads:

“From a tumour biology perspective, the main implication of our findings is that immunogenic MSI-H CRCs do not protect against release of CTCs, and its protective effect against metastatic disease is not by reducing CTCs. Finally, the main clinical implication of this study is on the clinical utility of CTCs for monitoring and surveillance with potentially differential baseline levels of CTCs associated with the two subtypes of CRC”

“Finally, these data suggest that immunogenic MSI-H CRCs do not suppress CTC release and that different reference ranges may be required for CTC enumeration for MSI-H and MSS CRC”

The authors should represent the CTC yield result in the tabular form. It will be helpful to understand the results clearly.

Table S1 has been added as supplementary material (lines 751-752).

The CTC numbers in the stage IV patient (Fig 2) is not conclusive or reliable as it was based on only one patient. The authors should represent the figures clearly. Please correct the legends of Figure 1. Please define MSI-H? Line 186….. there is no fig 5

For Figure 2: it has been made clear that this figure was based only on one patient (patient 16) and the Figure legend now reads:

“Figure 2. CTC number at different time points for patient S16 (Stage IV, MSS CRC).”

Figure 1 legend has been corrected and now reads:

“Figure 1. CTC number for Stage II (panel A and B), III (C and D) and I-III (panel E and F) MSS (left) vs MSI-H (right) colorectal cancer at different sample time points t1 (pre-operative), t2 (intra-operative), t3 (immediate post-operative) t4 (14-16 hrs post-operative).”

MSI-H was defined in the introduction (lines 56-60):

“It is known that CRC exhibiting high level MSI (MSI-H) are associated with increased tumour infiltrating lymphocytes (TILs) and is a marker of immunogenicity [1-3]. MSI-H CRCs are less likely to disseminate due to TILs as a protective factor, yet a double-edged sword exists in that MSI-H CRCs have more mutations and are associated with more adverse pathological features.”

Additional information on MSI has also been added to the introduction (see response to point 2 above)

References

Michael-Robinson JM, Biemer-Huttmann A, Purdie DM, Walsh MD, Simms LA, Biden KG, et al. Tumour infiltrating lymphocytes and apoptosis are independent features in colorectal cancer stratified according to microsatellite instability status. Gut. 2001;48(3):360-6. Kim JH, Kang GH. Molecular and prognostic heterogeneity of microsatellite-unstable colorectal cancer. World journal of gastroenterology : WJG. 2014;20(15):4230-43. Greenson JK, Bonner JD, Ben-Yzhak O, Cohen HI, Miselevich I, Resnick MB, et al. Phenotype of microsatellite unstable colorectal carcinomas: Well-differentiated and focally mucinous tumors and the absence of dirty necrosis correlate with microsatellite instability. The American journal of surgical pathology. 2003;27(5):563-70. Smyrk TC, Watson P, Kaul K, Lynch HT. Tumor-infiltrating lymphocytes are a marker for microsatellite instability in colorectal carcinoma. Cancer. 2001;91(12):2417-22. Tougeron D, Fauquembergue E, Rouquette A, Le Pessot F, Sesboue R, Laurent M, et al. Tumor-infiltrating lymphocytes in colorectal cancers with microsatellite instability are correlated with the number and spectrum of frameshift mutations. Modern pathology : an official journal of the United States and Canadian Academy of Pathology, Inc. 2009;22(9):1186-95. Popat S, Hubner R, Houlston RS. Systematic Review of Microsatellite Instability and Colorectal Cancer Prognosis. Journal of Clinical Oncology. 2005;23(3):609-18. Guastadisegni C, Colafranceschi M, Ottini L, Dogliotti E. Microsatellite instability as a marker of prognosis and response to therapy: a meta-analysis of colorectal cancer survival data. European journal of cancer (Oxford, England : 1990). 2010;46(15):2788-98. Kazama Y, Watanabe T, Kanazawa T, Tanaka J, Tanaka T, Nagawa H. Microsatellite instability in poorly differentiated adenocarcinomas of the colon and rectum: relationship to clinicopathological features. Journal of clinical pathology. 2007;60(6):701-4. Lamberti C, Lundin S, Bogdanow M, Pagenstecher C, Friedrichs N, Buttner R, et al. Microsatellite instability did not predict individual survival of unselected patients with colorectal cancer. International journal of colorectal disease. 2007;22(2):145-52. Buckowitz A, Knaebel HP, Benner A, Blaker H, Gebert J, Kienle P, et al. Microsatellite instability in colorectal cancer is associated with local lymphocyte infiltration and low frequency of distant metastases. British journal of cancer. 2005;92(9):1746-53. Toh J, Chapuis PH, Bokey L, Chan C, Spring KJ, Dent OF. Competing risks analysis of microsatellite instability as a prognostic factor in colorectal cancer. The British journal of surgery. 2017;104(9):1250-9.

Reviewer 2 Report

In the article by Toh et al authors described correlation between number of CTC cells and MSI in colorectal cancer to investigate prognostic relevance of CTC number. However, there are certain issues that should be addressed and clarified before publication.

T1, T2, T3 and T4: it might be confused by the T status of TNM; I would therefore suggest using t1, t2, t3 and t4. There are numerous typing errors. Manuscript needs English correction.

Materials and methods, CTC enrichment and enumeration Could you discuss, what would be the result, if you would depleted CD45 cells using immuno-magnetic CD45 beads? Results Clinical and surgical characteristics: there are repetitive sentences about high grade dysplasia. Figure 1: there is probably mistake in description Stage I-III MSS and MSI-H (A and B) groups (should be instead E and F). I would not recommend inclusion of the statistical significance between two and six patients; only the data on the overall number should be included (also this can be wrong estimation from only two patients). Poisson regression model: there is probably the wrong number of the Figure; it should be 4 instead of 5. Figure 4: the number of Stage should be replaced by numbers I, II, … to not be confused with time points. Discussion First paragraph: second sentence should be rephrased, it is too long and difficult to understand. The last two sentences are repetitive. Second paragraph: the last sentence is contradictory to Figure 3. Fourth paragraph: the term theory should be replaced with the hypothesis; the last sentence is to long and thus confusing, therefore should be rephrased. Fifth and sixth paragraph: beginning and middle part, respectively, are again repletion. Limitation of the study should be the beginning of the last paragraph and again in the middle there is to long sentence.

Author Response

The manuscript has been extensively edited for typographic and other errors and all changes have been indicated using track changes. The reviewers comments have been addressed below.

REVIEWER 2

In the article by Toh et al authors described correlation between number of CTC cells and MSI in colorectal cancer to investigate prognostic relevance of CTC number. However, there are certain issues that should be addressed and clarified before publication.

T1, T2, T3 and T4: it might be confused by the T status of TNM; I would therefore suggest using t1, t2, t3 and t4. There are numerous typing errors. Manuscript needs English correction.

All T1, T2, T3, T4 have been changed to t1, t2, t3, t4 throughout the manuscript.

Materials and methods, CTC enrichment and enumeration Could you discuss, what would be the result, if you would depleted CD45 cells using immuno-magnetic CD45 beads?

Depletion of CD45 cells has no relevance in the protocol as the Isoflux is a validated CTC enrichment system that significantly reduces the background of contaminating lymphocytes (CD45+). Adding an additional CD45+ depletion step would provide no gain and most likely result in reducing the number of CTCs captured, as with every additional enrichment step there is likely a small loss of rare CTCs.

Results Clinical and surgical characteristics: there are repetitive sentences about high grade dysplasia. Figure 1: there is probably mistake in description Stage I-III MSS and MSI-H (A and B) groups (should be instead E and F).

We have removed the repetitive sentences about high grade dysplasia and combined into one sentence. This now reads (lines 156-163): “

“Of the nineteen patients who had CTCs enumerated, two patients had high grade dysplasia without malignancy (these were 30 x 33 x 23mm and 57 x 50 x 55mm villous adenomas). Of the remaining 17 patients, 3 (17.6%) were Stage I, 6 (35.3%) Stage II, 7 (41.2%) Stage III and 1 (5.9%) Stage IV. 9 (52.9%) were right sided (caecal (n=4), ascending colon (n=4) and transverse colon (n=1), (41.2%) were left sided (rectum (n=4), rectosigmoid (n=1), sigmoid (n=3)). The histopathology of all seventeen patients were adenocarcinoma.

For Figure 1. The legend has been corrected and now reads:

Figure 1. CTC number for Stage II (panel A and B), III (C and D) and I-III (panel E and F) MSS (left) vs MSI-H (right) colorectal cancer at different sample time points t1 (pre-operative), t2 (intra-operative), t3 (immediate post-operative) t4 (14-16 hrs post-operative).”

I would not recommend inclusion of the statistical significance between two and six patients; only the data on the overall number should be included (also this can be wrong estimation from only two patients).

We made it clear throughout the text and also in the discussion that one of the limitations of this study was that we could not perform ‘insightful statistical analysis’ due to the low numbers. To help clarify, the individual patients, MSI group, Stage and CTC number will be included as a Table (Table S2) in the supplementary material so that the reader can clearly see this data. This was also one of the recommendations from Reviewer 1 (see response to Reviewer 1 point 6).

Poisson regression model: there is probably the wrong number of the Figure; it should be 4 instead of 5. Figure 4: the number of Stage should be replaced by numbers I, II, … to not be confused with time points.

Reference to Figure 5 in the text has been corrected to Figure 4 and we have replaced Stage 1, 2, 3, 4 to I-IV in the Figure as requested.

First paragraph: second sentence should be rephrased, it is too long and difficult to understand. The last two sentences are repetitive.

The last sentence has been deleted. The second sentence has been rephrased into two sentences (lines 437-440) and now reads:

“The evidence in the literature shows that CTCs may be important in prognostication of colon cancer [5] in predicting dissemination [6], overall and disease free survival [7] and lymph node involvement [8,9]. Higher levels of CTCs has been associated with worse outcome and may predict for poor disease free survival [10-12].”

Second paragraph: the last sentence is contradictory to Figure 3.

The last sentence is not contradictory to Figure 3. Figure 3 is specifically for right sided (panel A and panel B) and neoadjuvant therapy (panel C and panel D). Figure 1 should be referred to in this sentence and has been included.

The end of this paragraph has been changed for greater clarity and now reads (lines 446-451):

“However, notably, our study showed that MSI-H CRCs (which have been reported to be immunogenic and associated with enhanced survival) [3,4] were associated with increased peri-operative release of CTCs (Figure 1). Further, both the mean and median CTC count at all time points were higher in the MSI-H group compared to the MSS group. Our hypothesis that increased TILs would decrease peri-operative release of CTCs in MSI-H CRCs was not supported by the data.”

Fourth paragraph: the term theory should be replaced with the hypothesis; the last sentence is to long and thus confusing, therefore should be rephrased.

The term theory has been replaced with hypothesis. The last sentence has been rephrased and separated into two sentences and now reads (lines 519-522):

“One hypothesis is that CTCs released into the bloodstream of patients with MSI-H CRCs usually remain microsatellite unstable [16] which maintains an enhanced immunogenic response from circulating lymphocytes in the bloodstream [17]. Thus, its presence may not represent the same risk of metastases as CTCs associated with MSS CRCs [18,19].

Fifth and sixth paragraph: beginning and middle part, respectively, are again repletion. Limitation of the study should be the beginning of the last paragraph and again in the middle there is to long sentence.

We have removed some of the repetitive sentences and the too long sentence has been split into two sentences and the limitations moved to the start of the last paragraph and now reads (lines 524-590):

So what about the protective effect of MSI? The immunogenicity of MSI is believed to be associated with presence of TILs. Collinearity between MSI-H status and TILs has been shown in clinicopathological data, including the data from our own cohort study[20] and has been traditionally associated with a better prognosis [4,21]. It is believed that the survival and clinical benefits of MSI may be due to its immunogenicity, with MSI associated with increased TILs [2,22-25] and TILs associated with a better prognosis [2,26-35]. MSI-H CRCs may also be associated not only with an abundance of intra-tumoral TILs, but also with a higher density of associated cytotoxic, helper and regulatory T lymphocytes in peripheral blood [17] as well as increased activity in the bone marrow [36]. Studies have shown that both the immunogenic TILs response at the tumour site as well as circulatory system are believed to be associated with a decreased risk of lymph node metastases [19,37] and distant metastases [18] in MSI CRCs. The assumption, hence, would be that MSI-H CRCs should be associated with decreased CTC dissemination into the blood.

However, this study has shown that the immuno-biology of MSI-H CRCs is more complex than this. As is well established in the literature, MSI-H CRCs are also associated with poor differentiation [19,38] are larger and more likely to be mucinous [39-41], as well as being higher grade tumours with a greater mutational burden and the mucinous phenotype more likely to be associated with signet cells. It is also believed that MSI-H CRCs are associated with a high number frameshift mutation peptides (FSPs) [42,43] when compared to MSS CRCs [44,45]. The MSI hyper-mutational state is also thought to be the reason for poor differentiation and other adverse pathological features of the tumour. The higher grade of tumour associated with hyper-mutational state may be the reason for the increased peri-operative release of CTCs. On Poisson regression analysis, our study showed no difference in pre-operative (t1) CTC numbers, but a statistically significant difference in intra-operative (t2 and t3) and post-operative (t4) release of CTCs. From the literature, immunogenic TILs may reduce the risk of dissemination, but this study showed that it does not do so by preventing the release of CTCs. We currently do not have long term survival and recurrence data nor CTC measurements outside of the 24 hour window peri-operatively, but a study looking at recurrence, survival and CTC number at 6 months, 1 year, 3 year and 5 years would be interesting and would examine the role that CTCs may play in the post-operative management of CRCs including whether it may be used to accurately predict recurrence or survival.

Notably, this pilot study has shown results which warrants further investigations. The increased CTC positivity seen in MSI-H CRCs is a significant point of difference from the currently understood immune-biological mechanisms associated with MSI-H CRCs. In practice, this means that the benchmark cut-off points for CTC enumeration may be influenced by tumour characteristic, and future clinical applications of CTC in CRC management may need to take this into consideration. Another consideration is that the improved survival associated with immunogenic MSI-H CRCs may not be as profound as once believed, and the increased peri-operative release of CTCs in MSI-H CRCs may be revisited with future studies with a larger patient cohort. This could prove a useful follow up of the suggestions made by the European Society of Medical Oncology (ESMO) guidelines that patients with stage II colon cancer with MSI are at very low risk of recurrence and unlikely to benefit from chemotherapy.

In this study the overall patient number (n=20) was low and a limitation for insightful statistical analysis. Another consideration is that the improved survival associated with immunogenic MSI-H CRCs may not be as profound as once believed, and the increased peri-operative release of CTCs in MSI-H CRCs may be revisited with future studies with a larger patient cohort. Suggestions made by the European Society of Medical Oncology (ESMO) guidelines that patients with stage II colon cancer with MSI are at very low risk of recurrence and unlikely to benefit from chemotherapy may possibly be monitored using CTCs.

In this study the overall patient number (n=20) was low and a limitation for insightful statistical analysis.””

Round 2

Reviewer 2 Report

In the article by Toh et al authors described correlation between number of CTC cells and MSI in colorectal cancer to investigate prognostic relevance of CTC number. However, there are still two points that should be corrected before publication.

Materials and methods, CTC enrichment and enumeration Could you discuss, what would be the result, if you would depleted CD45 cells using immuno-magnetic CD45 beads?

Author response: ”Depletion of CD45 cells has no relevance in the protocol as the Isoflux is a validated CTC enrichment system that significantly reduces the background of contaminating lymphocytes (CD45+). Adding an additional CD45+ depletion step would provide no gain and most likely result in reducing the number of CTCs captured, as with every additional enrichment step there is likely a small loss of rare CTCs.”

Remaining question: I know that depletion of CD45 cells has no relevance in the described protocol; I was wondering whether you would expect different results using other tubes, e.g. from Norgen biotek, following depletion of CD45 cells and analyzing the remaining CTCs.

Clinical and surgical characteristics: there are repetitive sentences about high grade dysplasia.

“Of the nineteen patients who had CTCs enumerated, two patients had high grade dysplasia without malignancy.« This sentence is written twice.

I would not recommend inclusion of the statistical significance between two and six patients; only the data on the overall number should be included (also this can be wrong estimation from only two patients).

Author response: ”We made it clear throughout the text and also in the discussion that one of the limitations of this study was that we could not perform ‘insightful statistical analysis’ due to the low numbers. To help clarify, the individual patients, MSI group, Stage and CTC number will be included as a Table (Table S2) in the supplementary material so that the reader can clearly see this data. This was also one of the recommendations from Reviewer 1 (see response to Reviewer 1 point 6).

Remaining consideration: Within the manuscript, there are two paragraphs where statistical significance should be omitted. Any of the statistician would suggest you the same. I strongly agree, that this is the interesting result, however there are still ONLY TWO AND FOUR/SIX SAMPLES. I would therefore suggest replacing in the bellow paragraphs that the sample size was to small to perform the reliable statistical analysis.

“For this analysis there were no MSI-H patients in the Stage I and IV groups, however there were two MSI-H (caecum and ascending colon) patients in the Stage II group. Whereas there were four Stage II MSS CRC (caecum, rectosigmoid and two rectum) patients. For the Stage II patients there was a significant spike at the t2 timepoint for the MSI-H group (Figure 1; A and B)) which was statistically significant between the two groups.”

“There were eight left-sided CRCs in this study. There were no MSI-H CRC in this subgroup. Two patients received neoadjuvant therapy, whereas six patients did not receive neoadjuvant therapy. There were four rectal cancers, one rectosigmoid and three sigmoid cancers. The CTC number overall was low in this subgroup, with no statistically significant difference between those who received neoadjuvant therapy and those who did not (Figure 3, panels C and D).”

Author Response

Materials and methods, CTC enrichment and enumeration Could you discuss, what would be the result, if you would depleted CD45 cells using immuno-magnetic CD45 beads?

Remaining question: I know that depletion of CD45 cells has no relevance in the described protocol; I was wondering whether you would expect different results using other tubes, e.g. from Norgen biotek, following depletion of CD45 cells and analyzing the remaining CTCs.

This is an interesting hypothetical question. It is possible that with the use of preservative tubes and immuno-magnetic depletion of CD45 cells followed by CTC analysis this may result in additional CTCs being identified. However, I don’t believe this would alter the proportion of CTCs for the patients across time points or when analysed in relation to MSI status. In our case we used EDTA tubes and processed all samples within 24 hrs from blood draw. This was a strict requirement as it is known that CTC isolation efficiency is optimal up to 24hrs when using EDTA tubes and then tapers off. It is likely for future studies we would use preservative tubes (such as CellSave) to provide even greater confidence for optimal CTC capture efficiency.

Clinical and surgical characteristics: there are repetitive sentences about high grade dysplasia. “Of the nineteen patients who had CTCs enumerated, two patients had high grade dysplasia without malignancy.« This sentence is written twice.

Thank you for pointing this out. The duplicated sentence has now been removed.

Remaining consideration: Within the manuscript, there are two paragraphs where statistical significance should be omitted. Any of the statistician would suggest you the same. I strongly agree, that this is the interesting result, however there are still ONLY TWO AND FOUR/SIX SAMPLES. I would therefore suggest replacing in the bellow paragraphs that the sample size was to small to perform the reliable statistical analysis.

“For this analysis there were no MSI-H patients in the Stage I and IV groups, however there were two MSI-H (caecum and ascending colon) patients in the Stage II group. Whereas there were four Stage II MSS CRC (caecum, rectosigmoid and two rectum) patients. For the Stage II patients there was a significant spike at the t2 timepoint for the MSI-H group (Figure 1; A and B)) which was statistically significant between the two groups.”

In accordance with the recommendations of the reviewer this paragraph has now been changed to:

“For this analysis there were no MSI-H patients in the Stage I and IV groups, however there were two MSI-H (caecum and ascending colon) patients in the Stage II group. Whereas there were four Stage II MSS CRC (caecum, rectosigmoid and two rectum) patients. For the Stage II patients there was a significant spike at the t2 timepoint for the MSI-H group (Figure 1; panel A and panel B). However, the sample size was too small to perform reliable statistical analysis.”

“There were eight left-sided CRCs in this study. There were no MSI-H CRC in this subgroup. Two patients received neoadjuvant therapy, whereas six patients did not receive neoadjuvant therapy. There were four rectal cancers, one rectosigmoid and three sigmoid cancers. The CTC number overall was low in this subgroup, with no statistically significant difference between those who received neoadjuvant therapy and those who did not (Figure 3, panels C and D).”

In accordance with the recommendations of the reviewer this second paragraph has now been changed to:

"There were eight left-sided CRCs in this study of which none exhibited MSI. Two patients received neoadjuvant therapy, whereas six patients did not receive neoadjuvant therapy. There were four rectal cancers, one rectosigmoid and three sigmoid cancers. The CTC number overall was low in this subgroup, however it appeared there was a difference between those who received neoadjuvant therapy and those who did not (Figure 3, panels C and D). Reliable statistical analysis was not performed due to the small sample size."